health and disease and epidemiology

SARS-CoV-2, epidemic modelling, non-pharmacological intervention

**Author for correspondence:**
M. R. Deinert
e-mail: mdeinert@mines.edu

†The first two authors contributed equally to this work.

# Estimating the effect of non-pharmaceutical interventions on US SARS-CoV-2 infections in the first year of the pandemic

N. A. Duncan[1,†], G. F. L'Her[1,†], A. G. Osborne[1], S. L. Sawyer[2] and M. R. Deinert[1]

[1]Mechanical Engineering, The Colorado School of Mines, Golden, CO 10996, USA
[2]Molecular Biology, University of Colorado at Boulder, Boulder, CO, USA

MRD, 0000-0002-3951-5568

SARS-CoV-2 emerged in late 2019 as a zoonotic infection of humans, and proceeded to cause a worldwide pandemic of historic magnitude. Here, we use a simple epidemiological model and consider the full range of initial estimates from published studies for infection and recovery rates, seasonality, changes in mobility, the effectiveness of masks and the fraction of people wearing them. Monte Carlo simulations are used to simulate the progression of possible pandemics and we show a match for the real progression of the pandemic during 2020 with an $R^2$ of 0.91. The results show that the combination of masks and changes in mobility avoided approximately 248.3 million ($\sigma = 31.2$ million) infections in the US before vaccinations became available.

## 1. Introduction

The twentieth century saw the emergence of the Spanish flu, MERS, HIV, Ebola, swine flu, Lasa and Marburg fever, and many more. In 2002, SARS-CoV emerged and the latest global threat is now from SARS-CoV-2 [1]. Epidemiological models played an important role in guiding the response to the pandemic during its initial months. However, the wide range of predictions also caused confusion [2]. This was to be expected as important input parameters, like infection and recovery rates, are poorly quantified with emergent pathogens like SARS-CoV-2, as are the effects of non-pharmacological interventions. However, in times of urgency, uncertainty contributes to public fatigue and can affect policy responses [3]. By the middle

of 2020, enough data had come out to put ranges on many of the parameters that were affecting the progression of the pandemic. By coupling these data with epidemiological models, it is possible to see which combination best matches data on infection rates and also to estimate the effect of interventions.

Models for how infectious diseases move through populations began to appear in the early 1900s and were developed to understand the drivers for epidemics and the number of people who could be affected. The simplest of these are compartmental models which group a population into categories (e.g. susceptible, infected, recovered). The basic concept dates to at least 1908 and the work of Ronald Ross on malaria [4], and the common formulation to Kermack & McKendrick [5]. Here, the susceptible and infected groups are coupled by an infection rate, $\alpha$, and the infected and recovered groups by a recovery rate, $\beta$ [6]. Compartmental models are often fit to case incidence rates in the early part of epidemics to estimate infection and recovery rates and the reproduction number ($R_0$) for the disease. For a three-compartment SIR model, $R_0 = \alpha/\beta$ [6].

Any degree of complexity can in principle be encompassed by expanding the number of compartments, and this is often done to consider different transmission rates among age groups and locations [e.g. 7]. However, if the underlying parameters are poorly understood, simpler models whose inputs have been measured can sometimes give better results [8]. As knowledge increases, the number of compartments can also be expanded. For SARS-CoV-2, even a 'minimal' SIR model requires understanding infection and recovery rates, mortality and hospitalization rates, and the effects of seasonality and non-pharmacological interventions. A wide range of estimates exist for these parameters, what affects them, and assumptions that underlay simple compartmental models in general. These will be discussed briefly to inform the ranges used.

Compartmental models assume homogeneous mixing and uniform susceptibility of the population. Whether people are uniformly susceptible to SARS-CoV-2 remains poorly understood at this time. Work on SARS-CoV showed some antibody cross-reactivity with common human coronaviruses [9,10]. It is possible, and perhaps likely, that this would also be true for SARS-CoV-2 and could afford some protection or immunity. Other work has shown that there could be cross-reactive T-cell recognition between seasonal human coronaviruses and SARS-CoV-2 in up to 50% of people [11]. Studies that followed individuals who recovered from SARS-CoV showed measurable antibodies in half of recovered people at 4 years and over 90% at 2 years [12]. This suggests that some level of immunity could be conferred, though it may fade with time, which is similar to seasonal human coronaviruses.

Infection rates in a naive population are often estimated with data on the exponential growth rate of the pandemic and data on recovery times, $g = (R_0 - 1)/\gamma = \alpha - \beta$ [13], where $\gamma$ is the serial interval time. However, for SARS-CoV-2, estimates for the exponential growth from studies in the first four months of the pandemic varied considerably, as did those for the serial interval. The recovery and incubation period can be similarly difficult to estimate with wide uncertainty. Table 1 gives examples for epidemic parameters from studies published in the first several months of the SARS-CoV-2 pandemic.

Because clinical data show hospitalization rates to be age dependent, it is also important to understand the relative susceptibility of different age groups to SARS-CoV-2 and if transmission rates are age dependent. An early study on potential hospitalization rates in the USA [7] used a contact matrix [25] which shifts simulated infections to the younger part of the population. However, data subsequently published on 391 cases in the early phase of the Wuhan epidemic showed that age played no role in infection rates [26]. A population screening study in Iceland, published subsequently, showed that younger people (and children under 10 in particular) were less likely to be infected than were older segments of the population. Here, two screens were performed, one targeted at people considered high risk (symptomatic persons, those having travelled to high-risk areas etc.) and one targeted at the general population [27]. The targeted screen showed more uniform infection rates across age groups, but with children again showing lower infection rates. A study of 3712 people in Germany showed children to have a lower rate of infection than adults but to carry similar viral loads when they do [28]. However, these results were for a population seeking testing, which could have skewed the results. In both the Iceland and Germany studies, data were collected over a period in which social distancing measures were in place, and this could also have affected the results.

Common human coronaviruses are known to exhibit seasonality which can arise both from environmental effects on viruses themselves but also seasonal factors that affect population dynamics [29]. Several studies have looked at the possible effects of seasonal forcing on SARS-CoV-2 with estimates for the decrease in summer infection rates from their yearly peak being as high as 82% and as low as 22% [23,24], table 1. Understanding whether SARS-CoV-2 shares the seasonality of seasonal human coronaviruses is complicated by the fact that mobility was dramatically reduced during parts of 2020 and physical distancing became widespread, e.g. [30]. The same was true for mask wearing, which was shown by a recent meta-analysis to have a significant effect on transmission [22], table 1.

**Table 1.** Estimates for critical parameters for modelling the SARS-CoV-2 pandemic. Data on the growth rate, serial interval time, $R_0$, recovery and incubation period as well as the effect of mask wearing and seasonality from peer-reviewed studies published in the first six months of the pandemic. The 9.8-day recovery period listed is post-onset of symptoms, while the 7.5-day period is reported as the time to become non-infectious.

| parameter | units | estimated | 95% CI | reference |
|---|---|---|---|---|
| growth rate (g) | 1/day | 0.29 | 0.21–0.37 | [14] |
| | | 0.19 | 0.09–0.69 | [15] |
| | | 0.10 | 0.05–0.16 | [1] |
| serial interval ($\gamma$) | day | 7.5 | 5.3–19 | [1] |
| | | 5.8 | 4.8–6.8 | [16] |
| | | 4.6 | 3.5–5.9 | [17] |
| | | 4.0 | 3.53–4.39 | [18] |
| $R_0$ | — | 5.7 | 3.8–8.9 | [14] |
| | | 2.6 | 2.1–5.1 | [15] |
| | | 2.2 | 1.4–3.9 | [1] |
| recovery period | day | 9.8 | 8.5–21.8 | [19] |
| | | 7.5 | 5.0–15.2 | [15] |
| incubation period | day | 5.2 | 1.8–12.4 | [20] |
| | | 5.1 | 4.5–5.8 | [21] |
| relative risk of infection (mask versus no mask) | — | 0.34 | 0.26–0.45 | [22] |
| seasonality (min/max infection rate) | — | 0.54 | 0.18–0.78 | [23,24] |

**Table 2.** Proportion of infections leading to hospitalization in the USA [32]. Data ranges are for hospitalizations in the USA between 12 February and 16 March 2020.

| age category | percentage range for symptomatic infected individuals | | |
|---|---|---|---|
| | hospitalization (%) | intensive care unit (%) | fatality (%) |
| 0–19 years | 1.6–2.5 | 0 | 0 |
| 20–44 years | 14.3–20.8 | 2.0–4.2 | 0.1–0.2 |
| 45–54 years | 21.2–28.3 | 5.4–10.4 | 0.5–0.8 |
| 55–64 years | 20.5–30.1 | 4.7–11.2 | 1.4–2.6 |
| 65–74 years | 28.6–43.5 | 8.1–18.8 | 2.7–4.9 |
| 74–85 years | 30.5–58.7 | 10.5–31.0 | 4.3–10.5 |
| 85 years and above | 31.3–70.3 | 6.3–29.0 | 10.4–27.3 |

A critical issue with emergent diseases is the stress that they can place on medical personnel and hospitals. This has been seen in many regions where SARS-CoV-2 has high infection rates. Estimates from China indicated a median hospital stay for SARS-CoV-2 of 11 days (IQR 10–14) [31] with severe cases having a median of 13 days (IQR 11.5–17) and non-severe cases a median of 10 days (IQR 10–13). Estimates for the percentage of symptomatic SARS-CoV-2 cases requiring hospitalization, with critical care and mortality rates, are available for the USA [32] and other countries [1], though these data are probably skewed as they are based only on people seeking care. Actual case fatality, hospitalization and critical care requirements appear to vary geographically as well [33] but this could also be influenced by differences in testing levels. Table 2 gives ranges for hospitalization, critical care and fatality rate by age of symptomatic infected people in the USA for data available as of 16 March 2020 [32]. The proportion of hospitalized cases, critical conditions cases and fatality varies by age [34]. However, these rates were also a moving target and have fallen with time [35,36].

Because data on hospitalization and mortality were being reported on the basis of symptomatic cases, it is also important to understand what fraction of SARS-CoV-2 infections are asymptomatic. This has been estimated at approximately 18% for passengers on the Diamond Princess cruise ship [37] and as high as 78% for unconfirmed reports coming out of China [38]. A preliminary study on the municipality of Gangelt, Germany (population approx. 12 529) looked at a random sample of approximately 1000 people [38]. Blood tests and PCR were used to determine the fraction of people with active infections and those who were SARS-CoV-2 IgG positive. The results showed 2% of the sample with an active infection, but 14% who were IgG positive, suggesting previous infection [39]. A study of 215 obstetric patients in the USA presenting for delivery showed 33 testing positive for SARS-CoV-2 but with 87.9% of them being asymptomatic [40].

Infection rates are affected not only by the properties of the virus, and the host it infects, but also by contact rates between individuals and steps that they may take to limit exposure. Cellphone data released by Google showed a significant drop in population mobility during periods of 2020 relative to January of that year [30]. This can be combined with data from the US Bureau of Labor Statistics for where people spend their time [41] to scale infection rates if contact rates are assumed to scale with mobility. Similarly, survey results released by YouGov gave time-dependent data for the fraction of people in the USA as a whole who were wearing face masks when in public [42]. These data can be used in conjunction with data on mask efficacy obtained from meta-analyses [22] to further scale infection rates. Unfortunately, neither the Google mobility data, nor the data from YouGov or the Bureau of Labor Statistics report an associated uncertainty.

In the present work, we use a 'minimal' compartmental model along with Monte Carlo simulations to pick the combination of factors that best matches the observed infection rates in the USA during 2020. This is in turn used to estimate the effect that masks and changes in mobility had in reducing infections. Because of the wide uncertainty in many of the key input parameters, we use the simplest compartment model, SIR, to simulate the fraction of the US population infected as a function of time. Infection rates are scaled for seasonality, mobility reductions, fraction of the population wearing masks and mask efficacy. This helps avoid assuming information, such as disease applicable contact matrixes, which is often poorly established in the early phases of an emergent pathogen (e.g. [7]). We use Monte Carlo sampling to show the spread in the predicted outcomes for the fraction of the population infected as a function of time, hospital beds needed, and mortality rates with and without seasonal effects.

## 2. Methods

Data for the epidemic growth rate, and time to become non-infectious, were taken from the studies listed in table 1. Because of the symmetry of the confidence intervals, the 0.10/day and 0.29/day exponential growth rates, g, were assumed to follow a normal distribution with the standard deviations chosen to best fit the confidence intervals. A third exponential growth rate (mean 0.16, CI: 0.11–0.21) was computed by fitting an exponential to the data from [1]. These were used in conjunction with the recovery period to determine the possible distributions for the infection rate, $\alpha = g - \beta$. The 7.5-day recovery period, $\beta$, was used here because it was given as specific to the time to become non-infectious and because its distribution (normal) was specified [15]. The infection rate was assumed to be uniform across age groups in keeping with the results from a contact tracing study done on the early phase of the Wuhan epidemic [26]. The infectious period was assumed to overlap with the incubation period to be consistent with observations of asymptomatic infection [16,43]. It was assumed that reinfection is not possible during the one year simulated time frame, thus played a minimal role [12]. The effect of seasonality was modelled using the range of estimates for seasonal forcing on infection rates [23,24]. We assume uniform susceptibility regardless of previous infection with seasonal corona viruses.

Reductions in mobility were used to scale infection rates by considering how much less time was spent by the US population in public (i.e. in transit, retail, grocery and workplace) relative to their January 2020 baseline values [30]. The effect of mask wearing was taken into account by considering the relative risk of infection, table 1, and survey data for the fraction of the US population that said they wore masks in public [42]. Data on mask efficacy were taken from a meta-analysis of 26 studies (Chu *et al.* [22], appendix 6). In 19 of these, the efficacy of the masks was determined in a setting where they were worn with physical distancing greater than or equal to 1 m (5 with social distancing of 2 m, 2 with 1.5–1.8 m and 12 with distancing of 1 m). As a result, we assume that the mask efficacy already factors in the effect of physical distancing and that the same people who are wearing masks

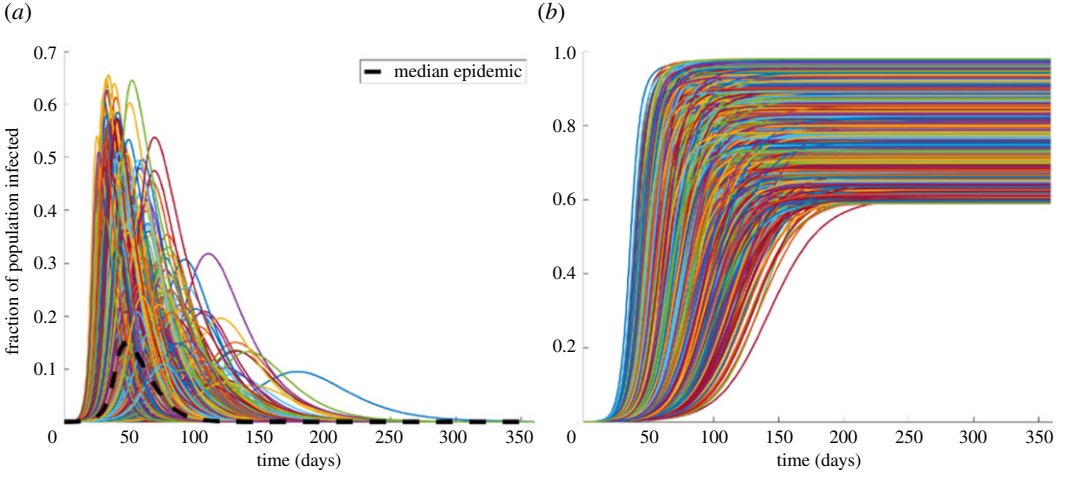

**Figure 1.** Monte Carlo simulation of the range of possible infection and recovery rates. (*a*) The results from 1000 iterations randomly sampling distributions for the infection rate and recovery rates. Depending on the combination of $\alpha$ and $\beta$ the peak can vary by more than a factor of 10. (*b*) The cumulative fraction of the population that would become infected. The results assume no social distancing.

are also maintaining physical distance from others as well. We sample from a uniform distribution for the range given in appendix 6 of Chu *et al.* [22].

Monte Carlo simulations for the SIR equations were used to randomly sample the infection and recovery rates and account for mobility, seasonality, and effect of mask wearing. The portion of a population that would be hospitalized at any one time was determined using the fraction of the infected population that would be symptomatic and the fraction of these that would require hospitalization or ICU care, as well as their residence time in each. For each of these, a uniform distribution was assumed due to the absence of other information. Actual SARS-CoV-2 infection rates in the USA were difficult to estimate in the early part of the pandemic because testing was limited. However, data on SARS-CoV-2 fatalities were more accurate. Here, we estimated infection rates using data on fatalities, combined with the average fatality rate from table 2, and testing data from [32]. The higher of the two was then used. A 7-day moving average was used to smooth noise and inconsistencies in reporting dates [44]. A complete description of the implementation is given in electronic supplementary material, note S1.

## 3. Results and discussion

Figure 1 shows the results of 1000 Monte Carlo simulations for the fraction of the US population that would be infected as a function of time (left) in the absence of any interventions. The horizontal axis is days after the appearance of the first case, which is thought to be 20 January for the USA [45]. Depending on the combination of $\alpha$ and $\beta$, the peak fraction of the population infected can vary by more than a factor of 6. Figure 1*b* shows the corresponding fraction of the population that would become infected before the epidemic burns itself out, which again varies considerably. The results show that in the absence of interventions a very high fraction of the population could be infected by SARS-CoV-2 across the full range of simulated epidemics.

When all the simulations in figure 1*a* are combined with the distribution of symptomatic people and the age-dependent distributions for mortality rates shown in table 2, the result can be used to determine the fraction of combinations which give a specific mortality rate for the US population. This is shown in figure 2. Here, the output of these combinations was filtered to remove the simulated epidemics producing the 2.5% highest and lowest peak mortality rates. Figure 2 then shows the range for the fraction of a population that would be expected to die due to SARS-CoV-2 as a function of time along with the epidemic that gives 97.5th percentile, 50th and 2.5th percentile mortality rates. All other possibilities between these limits sit within the shaded boundary. A heat map for their density is given in electronic supplementary material, note S1. The same method is applied to the hospitalization rate, with the output of the combination of Monte Carlo simulation with the symptomatic distribution and the age-specific hospitalization distributions filtered to remove the 2.5% highest and lowest hospitalization rates. This is shown in electronic supplementary material, note S1, figure S3. The corresponding heat map is given in electronic supplementary material, note S1. These

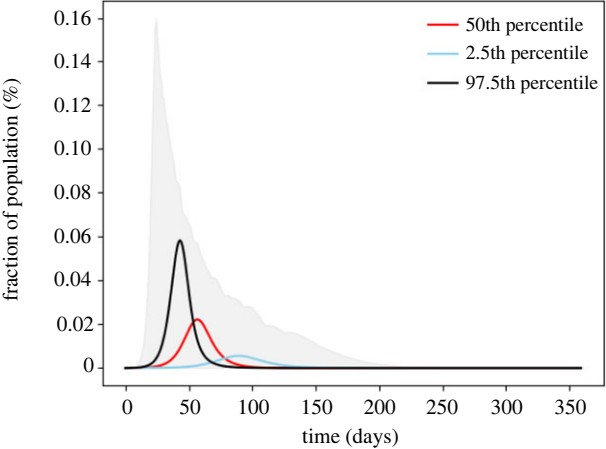

**Figure 2.** Monte Carlo simulation of the range of possible fatality rates. The daily fatalities result from $10^3$ simulations randomly sampling distributions for the infection rate, recovery rate which are inputs to $10^3$ simulations sampling the symptomatic rate and fatality rate. The results assume no non-pharmaceutical interventions. The grey shaded area is the envelope of all simulation results.

results show that, in the absence of interventions, low-impact epidemics are a small subset of the predicted outcomes.

As with the rates of infection and total population infected, the Monte Carlo simulations show a wide range of possible outcomes. The uncertainty is large with mortality rates falling in a range more than an order of magnitude wide. However, at the low end, one would expect mortality of 450 000 people in the USA, and at the upper end more than 4.4 million with a median of 2.0 million. The lower end of this range was passed in the USA in early February 2021 [33]. The median is just below the total number of fatalities seen in the USA from the Spanish flu in the 1918–1919 flu if the US population were scaled to its current level. This range of potential fatalities supports the serious social distancing and masks mandates put in place by many US states.

The results in figure 2 and electronic supplementary material, figure S3 are consistent with the ranges from early predictions for the pandemic, e.g. [46]. However, none of the simulation results shown in figures 1 and 2 are consistent with the way in which the pandemic unfolded in the USA. Seasonality effects, reductions in mobility, the use of facemasks and physical distancing all played a role in reducing infections and downstream hospitalizations and fatalities. When Monte Carlo simulations factor in the effect that these have on infection rates, the results are much more drawn out. The best match to the US pandemic through December 2020 is shown in figure 3a and the relative effects of mobility and mobility + mask wearing are shown along with the projected pandemic had these measures not been taken (figure 3b). The full range of Monte Carlo results is shown in electronic supplementary material, figure S7. In these simulations both mobility and mask wearing are seen to have a significant effect on the progression of the pandemic, with the latter dominating. While the best match of the Monte Carlo simulations to actual data is not perfect, ($R^2 = 0.91$), the results suggest that mask wearing and mobility reductions avoided approximately 248.3 million ($\sigma = 31.2$ million) infections in the USA before vaccinations began in December of 2020. Had a vaccination not become available, these non-pharmacological interventions would have reduced the total number of people infected before herd immunity was achieved. This effect is illustrated in electronic supplementary material, note S2, figure S10.

The effect of seasonality on the progression of the epidemic is important and a factor in the resurgence of SARS-CoV-2 infections in the autumn of 2020, figure 3a. However, it is also notable that a second peak in infections occurred in the summer of 2020, which is when seasonality would have predicted a minimum. This might suggest that SARS-CoV-2 is not as seasonal as other seasonal human corona viruses. However, these results should be interpreted with care. Immunization and infection will create population immunity, but immune naive children will always be present. How long immunity lasts is still an open question, and this could contribute to SARS-CoV-2 susceptible population and set up the circumstances for seasonally oscillating infections in the future (e.g. [15]).

The modelling approach used here has several limitations. Because distributions were not given for all of the values in table 1, only a subset of these data could be used. In particular, an average infection rate distribution was generated using the distributions for three estimates of the exponential growth for the epidemic in Wuhan and the single estimated recovery period, 7.5 days, for which a distribution was given (electronic supplementary material, Information note S1). However, it could also be that the

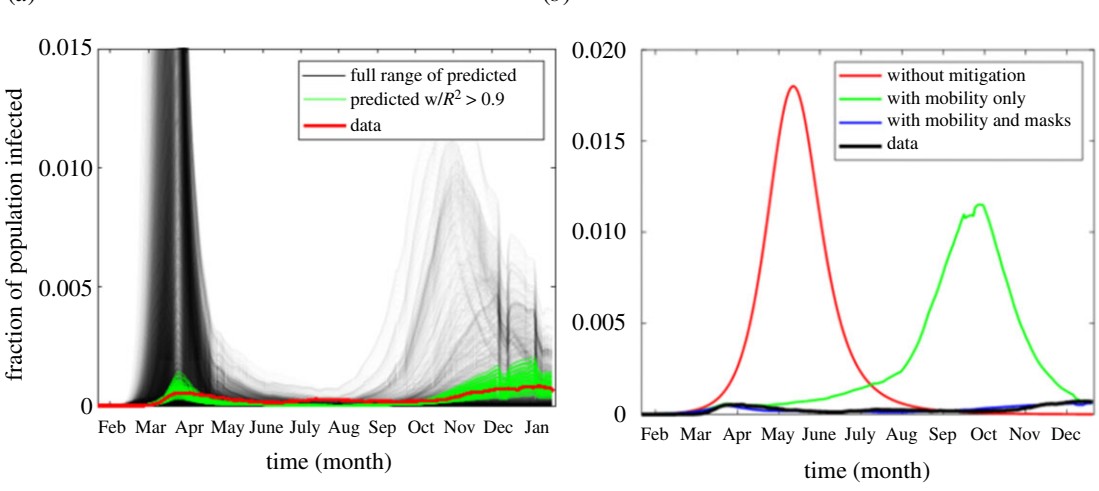

**Figure 3.** Effect of mobility and mask wearing on the time course of the US epidemic. (*a*) The progression of infections in the USA, red line, along with the best fits (green) of the Monte Carlo simulations when reductions in mobility and mask wearing are taken into consideration. The $R^2$ between the highlighted green simulations and actual data was greater than 0.90 and time runs from the day of appearance of the first case to one. (*b*) The best-fit simulation (blue) and the relative effects of mobility reductions and mobility + mask wearing, along with the predicted pandemic had these measures not been taken. The best match was with mobility + mask wearing.

different exponential growth rates correspond to different potential trajectories for the early SARS-CoV-2 epidemic. In this case, it would be more appropriate to sample three separate distributions for the infection rate instead of an average one, and this would result in a slightly wider range of possible outcomes. The results for this are shown in electronic supplementary material, note 3. The modelling approach used here makes no distinction between the ability of symptomatic and asymptomatic people to cause infection and uses an aggregate infection rate for the susceptible class as a whole. Other studies have used network population models, and estimates for the number of mild or asymptomatic cases, and concluded that people in this category have lower infection rates than do symptomatic people but still constitute 46–62% of total transmissions [47]. Data published subsequently showed that people become most infectious just before they become symptomatic and also that 46–55% of transmissions occurred in this time frame [16]. The assumption of age-independent infection rates used here is consistent with the contact tracing study done on the early phase of the Wuhan epidemic [26]. While it is well established that children and young people experience less severe disease as a population, the age dependence of susceptibility remains less clear. The effect of mobility reduction was based on a single time-dependent value from Google [30] and data from the US Bureau of Labor Statistics for where people spend their time [41]. However, no uncertainty was given for these data and so could not be factored into the modelling. It is important to note that there could be scaling effects that reduce infection rates for SARS-CoV-2 as urban population sizes decrease. This effect has been shown to occur with flu [48] and its potential influence on SARS-CoV-2 requires further investigation. The application of an SIR model to the population of the USA also has its limitations, as the population cannot be assumed to be completely mixed, and differences in climate could also mean that people are not uniformly susceptible. The way that infection rates were inferred from mortality data probably results in an underestimate. It is known that fatality rates decreased as medical interventions improved [35,36]. However, the temporal aspect is poorly quantified and was not considered as a result and this probably resulted in an underestimate of the infections. A fixed time from infection to death was assumed while time to mortality actually follows a distribution. On average, this does not strongly affect the timing of predicted infections given both the inconsistencies in reporting for the date of mortality [44] and the 7-day moving average used here for infections. This assumption was tested by using a measured distribution for the time from infection to death [49], implemented without the 7-day moving average for infections. The predicted number of infections avoided changed to 249.9 million with a standard deviation of 29.8 with a negligible change in $R^2$. Seasonality was also assumed to change in magnitude, but not in timing. However, the data show that the time when infection rates begin to rise varied by coronavirus strain by as much as +/− one month. This could affect the best fit of the model to the data. Finally, the possible effect of variants that arose in 2020 is not considered.

Despite the limitations described above, many of the simulations shown in figure 3 are remarkably close to what unfolded in 2020 and the best result has an $R^2$ of 0.91. Robert May observed that it does not make sense to add complexity to a model if key parameters can only be guessed at [8]. The present model considers infection and recovery rates, mobility reductions, the effectiveness of masks and seasonality. However, even with this relative simplicity one has to consider the results in context. The number of avoided infections came from the combination of parameters for which the resulting simulation most closely followed the infection profile of the actual pandemic. John von Neumann is said to have wryly noted that with four parameters he could fit an elephant, with five he could make it wiggle its trunk [50]. However, the parameters here are neither free nor arbitrary. Instead, they have a physical basis in affecting disease transmission and are varied over credible ranges for SARS-CoV-2, or related coronaviruses, with two fixed because no range was given (mobility and fraction of population wearing masks).

# 4. Conclusion

The first year of the SARS-CoV-2 pandemic saw a flood of studies that advanced the state of knowledge for this pathogen at a truly impressive rate. However, as table 1 shows, the range of values for the reproductive number, exponential growth rates, incubation and recovery periods, as well as mask effectiveness, are large. This can be exploited to understand important properties of both the virus and the possible effect of interventions. The results presented here show that for the range of infection and recovery rates for the Wuhan strain of SARS-CoV-2, the non-pharmacological measures taken avoided approximately 248.3 ($\sigma = 31.2$ million) million infections before a vaccine became available.

Key questions remain about the relationship between age and susceptibility to this virus, the effect of population scaling on infection rates, the lifetime of antibody response and the possibility that previous infection by human coronaviruses might afford protection. This is particularly true as new variants arise. As was done here, uncertainty can be used to understand whether properties of a virus exist (e.g. seasonality) and the effectiveness of interventions (e.g. social distancing). However, care should be taken in using complicated models to make explicit predictions when their input data do not exist or cannot be accurately inferred. Simple models, as was shown here, can do remarkably well.

Data accessibility. Data used in this study can be found in: CDC COVID-19 Response Team *et al.*, 'Severe Outcomes Among Patients with Coronavirus Disease 2019 (COVID-19) 'United States, 12 February–16 March 2020', MMWR Morb. Mortal. Wkly. Rep., vol. 69, no. 12, pp. 343–346, March 2020, doi:10.15585/mmwr.mm6912e2. Coronavirus Resource Center, Johns Hopkins University of Medicine. https://coronavirus.jhu.edu/ (accessed 1 April 2020). COVID-19 Community Mobility Report, COVID-19 Community Mobility Report. https://www.google.com/covid19/mobility?hl=en (accessed 1 April 2021). Personal measures taken to avoid COVID-19 | YouGov, https://today.yougov.com/topics/international/articles-reports/2020/03/17/personal-measurestaken-avoid-covid-19 (accessed 1 April 2021). United States COVID-19 Cases and Deaths by State over Time | Data | Centers for Disease Control and Prevention. https://data.cdc.gov/Case-Surveillance/United-States-COVID-19-Cases-and-Deathsby-State-o/9mfq-cb36.

Electronic supplementary material is available online [51].

Authors' contributions. N.A.D.: data curation, formal analysis, investigation, methodology, software, validation, visualization, writing—original draft, writing—review and editing; G.F.L.: data curation, formal analysis, investigation, methodology, software, validation, visualization, writing—original draft, writing—review and editing; A.G.O.: methodology; S.L.S.: writing—original draft; M.R.D.: conceptualization, formal analysis, investigation, methodology, project administration, supervision, validation, visualization, writing—original draft, writing—review and editing.

All authors gave final approval for publication and agreed to be held accountable for the work performed therein.

Conflict of interest declaration. We declare we have no competing interests.

Funding. We received no funding for this study.

Acknowledgments. Special thanks to Richard Garwin for a discussion of elephants and things that might fit them.

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
