## [Peer Review File · Royal Society Open Science]

Review History

RSOS-210875.R0 (Original submission)

Review form: Reviewer 1

Is the manuscript scientifically sound in its present form?

Yes

Are the interpretations and conclusions justified by the results?

No

Is the language acceptable?

Yes

Do you have any ethical concerns with this paper?

No

Have you any concerns about statistical analyses in this paper?

Yes

Recommendation?

Major revision is needed (please make suggestions in comments)

Comments to the Author(s)

Please find comments for the authors in the attached file (see Appendix A).

Decision letter (RSOS-210875.R0)

Dear Dr Deinert

The Editors assigned to your paper RSOS-210875 "Non-Pharmaceutical Interventions Prevented ~250 Million United States SARS-CoV-2 Infections in 2020" have now received comments from reviewers and would like you to revise the paper in accordance with the reviewer comments and any comments from the Editors. Please note this decision does not guarantee eventual acceptance.

Please submit your revised manuscript and required files (see below) no later than 21 days from today's (ie 10-Sep-2021) date. Note: the ScholarOne system will 'lock' if submission of the revision is attempted 21 or more days after the deadline. If you do not think you will be able to meet this deadline please contact the editorial office immediately.

Kind regards,

Royal Society Open Science Editorial Office
Royal Society Open Science

on behalf of Professor Chris Budd (Associate Editor) and Nick Pearce (Subject Editor)
openscience@royalsociety.org

Associate Editor Comments to Author (Professor Chris Budd):

Associate Editor: 1

Comments to the Author:

This reviewer of paper has expressed concerns that the interpretations and conclusions of this paper are not justified by the results, and in particular the evaluation of the simple SIR model used. They are concerned about the quantification of the uncertainty in the model given the various types of NPIs used. These are important observations. I suggest that the authors make a major review of their paper in order to bring out, and to quantify, the uncertainty in the models that they have used. This discussion should also include a much more careful structuring of the description of the way that the various parameters in the model are used. I hope that they can do this, as the headline result of the paper is certainly of interest.

Reviewer comments to Author:

Reviewer: 1

Comments to the Author(s)

Please find comments for the authors in the attached file.

===PREPARING YOUR MANUSCRIPT===

===PREPARING YOUR REVISION IN SCHOLARONE===

Author's Response to Decision Letter for (RSOS-210875.R0)

See Appendix B.

RSOS-210875.R0 (Revision)

Review form: Reviewer 1

Is the manuscript scientifically sound in its present form?

Yes

Are the interpretations and conclusions justified by the results?

No

Is the language acceptable?

No

Do you have any ethical concerns with this paper?

No

Have you any concerns about statistical analyses in this paper?

No

Recommendation?

Accept with minor revision (please list in comments)

Comments to the Author(s)

Please see attached file (Appendix C).

Decision letter (RSOS-210875.R1)

Dear Dr Deinert

On behalf of the Editors, we are pleased to inform you that your Manuscript RSOS-210875.R1 "Non-Pharmaceutical Interventions Prevented ~250 Million United States SARS-CoV-2 Infections in 2020" has been accepted for publication in Royal Society Open Science subject to minor

revision in accordance with the referees' reports. Please find the referees' comments along with any feedback from the Editors below my signature.

Please submit your revised manuscript and required files (see below) no later than 7 days from today's (ie 25-Mar-2022) date. Note: the ScholarOne system will 'lock' if submission of the revision is attempted 7 or more days after the deadline. If you do not think you will be able to meet this deadline please contact the editorial office immediately.

on behalf of Professor Chris Budd (Associate Editor) and Nick Pearce (Subject Editor)
openscience@royalsociety.org

Associate Editor Comments to Author (Professor Chris Budd):

Associate Editor: 1

Comments to the Author:

The reviewer has made a comprehensive list of minor suggestions to improve the paper. These are mostly along the lines of adding in a few extra points of justification of the models that you are using. If you can address these then I am happy to accept the paper.

Reviewer comments to Author:

Reviewer: 1

Comments to the Author(s)

Please see attached file.

===PREPARING YOUR MANUSCRIPT===

one version should clearly identify all the changes that have been made (for instance, in coloured highlight, in bold text, or tracked changes);
 a 'clean' version of the new manuscript that incorporates the changes made, but does not highlight them. This version will be used for typesetting.

===PREPARING YOUR REVISION IN SCHOLARONE===

- An individual file of each figure (EPS or print-quality PDF preferred [either format should be produced directly from original creation package], or original software format).
 - An editable file of each table (.doc, .docx, .xls, .xlsx, or .csv).
 - An editable file of all figure and table captions.
- Note: you may upload the figure, table, and caption files in a single Zip folder.
- Any electronic supplementary material (ESM).
 - If you are requesting a discretionary waiver for the article processing charge, the waiver form must be included at this step.
 - If you are providing image files for potential cover images, please upload these at this step, and inform the editorial office you have done so. You must hold the copyright to any image provided.
 - A copy of your point-by-point response to referees and Editors. This will expedite the preparation of your proof.

- Ensure that your data access statement meets the requirements at <https://royalsociety.org/journals/authors/author-guidelines/#data>><https://royalsociety.org/journals/authors/author-guidelines/#data>. You should ensure that you cite the dataset in your reference list. If you have deposited data etc in the Dryad repository, please only include the 'For publication' link at this stage. You should remove the 'For review' link.
- If you are requesting an article processing charge waiver, you must select the relevant waiver option (if requesting a discretionary waiver, the form should have been uploaded, see 'File upload' above).
- If you have uploaded any electronic supplementary (ESM) files, please ensure you follow the guidance at <https://royalsociety.org/journals/authors/author-guidelines/#supplementary-material> to include a suitable title and informative caption. An example of appropriate titling and captioning may be found at [https://figshare.com/articles/Table_S2_from_Is_there_a_trade-off_between_peak_performance_and_performance_breadth_across_temperatures_for_aerobic_sc](https://figshare.com/articles/Table_S2_from_Is_there_a_trade-off_between_peak_performance_and_performance_breadth_across_temperatures_for_aerobic_scope_in_teleost_fishes_/3843624) ope_in_teleost_fishes_/3843624.

Author's Response to Decision Letter for (RSOS-210875.R1)

See Appendix D.

Decision letter (RSOS-210875.R2)

Dear Dr Deinert:

I am pleased to inform you that your manuscript entitled "Estimating the effect of non-pharmaceutical interventions on US SARS-CoV-2 infections in the first year of the pandemic" is now accepted for publication in Royal Society Open Science.

Please send to the office the original files for the figures and tables included in your manuscript - without these, the processing of your proof may be delayed.

In addition, we note that 'glher@mymail.mines.edu' is not currently able to receive messages from the journal. We require that all authors have an active email address able to receive correspondence. Please can you ensure that your colleague either 'white lists' correspondence from Manuscript Central ScholarOne, or provides the journal team with an active alternative email address?

Please remember to make any data sets or code libraries 'live' prior to publication, and update any links as needed when you receive a proof to check - for instance, from a private 'for review' URL to a publicly accessible 'for publication' URL. It is also good practice to add data sets, code and other digital materials to your reference list.

COVID-19 rapid publication process:

We are taking steps to expedite the publication of research relevant to the pandemic. If you wish, you can opt to have your paper published as soon as it is ready, rather than waiting for it to be published the scheduled Wednesday.

This means your paper will not be included in the weekly media round-up which the Society sends to journalists ahead of publication. However, it will still appear in the COVID-19 Publishing Collection which journalists will be directed to each week (<https://royalsocietypublishing.org/topic/special-collections/novel-coronavirus-outbreak>).

If you wish to have your paper considered for immediate publication, or to discuss further, please notify openscience_proofs@royalsociety.org and press@royalsociety.org when you respond to this email.

Royal Society Open Science is a fully open access journal. A payment may be due before your article is published. Our partner Copyright Clearance Centre will contact the corresponding author about your open access options (if you have any queries regarding fees, please see <https://royalsocietypublishing.org/rsos/charges> or contact authorfees@royalsociety.org).

on behalf of Professor Chris Budd (Associate Editor) and Professor Nick Pearce (Subject Editor).

Associate Editor Professor Chris Budd Comments to Author:

<https://www.facebook.com/RoyalSocietyPublishing/>

Appendix A

In the paper “Non-Pharmaceutical Interventions Prevented ~250 Million United States SARS-CoV-2 Infections in 2020” the authors explore SARS-CoV-2 epidemic projections for the US using a simple SIR model given a range of plausible transmission parameters and age-dependent estimates for severe outcomes available in mid 2020. Projected trajectories for the year following January 2020, that incorporate measured time-dependent mobility and mask wearing behaviour into the SIR model, are compared to observed case, hospitalisation and case data using R^2 as a measure of model performance. The authors estimate averted infections in the year following pandemic emergence due to social distancing and mask wearing, and note that including seasonality, mask-wearing and social distancing provides the best fit to the available data.

My main comment is that given the headline result is the number of averted infections due to NPIs prior to the availability of vaccines, it is worth thinking about trying to quantify the uncertainty for this prediction within the model framework presented. As written the authors present a single best fitting model based on the variance explained in each data stream, but the range of estimates for similarly reasonable fits is not discussed as far as I can see despite parameter uncertainty being a significant theme in the first stage of the paper. As part of this it could be worth exploring uncertainty in the effect of mask wearing and social mobility, as I understand these were fixed in the Monte Carlo simulations. I think this is important for both the mobility data (since for example this is not providing information about interactions in private households) and the mask wearing term (since in the model this seems like it would represent all micro-distancing behaviours including staying 1-2 metres away, hand washing etc).

Minor comments:

I find the structure of the introduction a little confusing, given a lot of focus on things that probably should be included in the model but aren't (like heterogeneities in susceptibility and asymptomatic cases) and the introduction of some of the main data sources. Structuring the introduction to separately discuss parameters available early on that enabled projection and subsequent data that enables evaluation of NPIs could be helpful.

Is Table 2 copied directly from CDC COVID-19 Response Team et al., “Severe Outcomes Among Patients with Coronavirus Disease 2019 (COVID-19) — United States, February 12–March 16, 2020,” *MMWR Morb. Mortal. Wkly. Rep.*, vol. 69, no. 12, pp. 343–346, Mar. 2020, doi: 10.15585/mmwr.mm6912e2?. If so could this be better placed in the supplementary index?

The authors mention homogeneity of their model population as a limitation, which I agree is a significant limitation of this approach. However, the discussion about reasons for potential variation in susceptibility focusses on the existence of cross-reactive T-cells from seasonal coronavirus exposure. Mentioning the age-dependence of susceptibility inferred in other modelling works (e.g. Davies et al., <https://www.nature.com/articles/s41591-020-0962-9>) is probably warranted here.

In line 61 the use of broad is unclear: does broad refer to wide uncertainty in the value or widely varying between hosts (or both?).

Line 94 discussion of study in reference 31 is unclear, were participants tested longitudinally, otherwise how does this constrain the asymptomatic fraction?

I think the method that has been adopted to infer cases has some limitations not mentioned. When inferring infections from deaths this is sensitive to the distribution of delays from infection to death, but the authors have assumed a fixed delay. A further limitation is that even if testing was

abundant, changing restrictions may alter the age-distribution of cases and result in time-varying under-ascertainment.

Related to my major comment: from figure S4 it looks like the phase of the seasonality term has also been fixed and may plausibly vary by weeks-months. It could be worth including this effect in a sensitivity analysis.

SI line 19 - slightly confusing to refer to the birth rate as the growth rate when the later has another meaning in the paper.

SI line 100 - Mobility data doesn't track social interactions in households which are potential a large source of transmission events, may be worth noting as another limitation.

Figure S7 - serial interval shown in Table 1. Although the serial interval and generation time interval distributions should have the same mean, they are not the same so it is a bit confusing to use the terms interchangeably.

Figure S7 - it may be my copy, but I can't easily make out the best fit in this figure, perhaps consider showing other fits as a shaded envelope?

Supplementary Note 2: this is for NPIs remaining at constant rate indefinitely? It could be worth clarifying this in the discussion.

Appendix B

Reviewer(s)' Comments to Author:

Referee: 1

Comments to the Author

My main comment is that given the headline result is the number of averted infections due to NPIs prior to the availability of vaccines, it is worth thinking about trying to quantify the uncertainty for this prediction within the model framework presented. As written the authors present a single best fitting model based on the variance explained in each data stream, but the range of estimates for similarly reasonable fits is not discussed as far as I can see despite parameter uncertainty being a significant theme in the first stage of the paper. As part of this it could be worth exploring uncertainty in the effect of mask wearing and social mobility, as I understand these were fixed in the Monte Carlo simulations. I think this is important for both the mobility data (since for example this is not providing information about interactions in private households) and the mask wearing term (since in the model this seems like it would represent all micro-distancing behaviours including staying 1-2 metres away, hand washing etc).

We thank the author for these comments and the resulting modifications greatly enhance the paper. We previously showed only the best matching simulation for distributions in infection rate, recovery rate, and seasonality, with mobility changes, fraction of the population wearing masks, and the efficacy of masks fixed. In the updated manuscript we have added in a distribution for mask efficacy though mobility and the fraction of the population that wears masks remains fixed (no uncertainty is given for these data). In the revised manuscript we show all simulations with an $R^2 > 0.9$ (Fig. 3, green, left figure). We use this to estimate the uncertainty in the main result, 250 million avoided infections, and find it to be +/- 30 million. Additionally, we added a Fig. S8 (below) to the SI material that shows the narrowed ranges of the infection and recovery rates that resulted in the $R^2 > 0.9$.

Figure 3. Effect of mobility and mask wearing on the time course of the US epidemic. (left) the progression of infections in the US, red line, along with the best fits (green) of the Monte Carlo simulations when reductions in mobility and mask wearing are taken into consideration. The R^2 between the highlighted green simulations and actual data was greater than 0.90 and time runs from the day of appearance of the first case to one. (right) The best fit simulation (blue) and the relative effects of mobility reductions, and mobility + mask wearing along with the predicted pandemic had these measures not been taken. The best match was with mobility + mask wearing.

The data on mask efficacy that we use was drawn from 26 studies (Appendix 6 - D. K. Chu *et al.*,

Figure S8. Infection (left) and Recovery (right) rate distributions that resulted in an $R^2 > 0.9$. The original distribution shown in black along with those simulations (green) with NPI that resulted in an $R^2 > 0.9$ to the actual infection data within the US.

“Physical distancing, face masks, and eye protection to prevent person-to-person transmission of SARS-CoV-2 and COVID-19: a systematic review and meta-analysis,” *The Lancet*, 395, 10242, pp. 1973–1987, Jun. 2020, doi: 10.1016/S0140-6736(20)31142-9). In 19 of these, the efficacy of the masks was determined in a setting where they were worn with physical distancing of ≥ 1 . As a result, we assume that the mask efficacy already factors in the effect of social distancing. To make this point clear, we have added the following to the methods section:

“Data on mask efficacy was taken from a meta-analysis of 26 studies (Appendix 6 - Chu *et al.*, [23]). In 19 of these the efficacy of the masks was determined in a setting where they were worn with physical distancing ≥ 1 (5 with social distancing of 2m, 2 with 1.5-1.8m, and 12 with distancing of 1m). As a result, we assume that the mask efficacy already factors in the effect of physical distancing and that the same people who are wearing masks are also maintaining physical distance from others as well, and vice versa. We sample from a uniform distribution for the range given in Appendix 6 of Chu *et al* (2020) [23].”

I find the structure of the introduction a little confusing, given a lot of focus on things that probably should be included in the model but aren't (like heterogeneities in susceptibility and asymptomatic cases) and the introduction of some of the main data sources. Structuring the introduction to separately discuss parameters available early on that enabled projection and subsequent data that enables evaluation of NPIs could be helpful.

The point we are making with the extensive discussion of the “...things that probably should be included in the model but aren't...” is that there is so much uncertainty that including them is not justified. Even now, a year after we first submitted this paper, there is still so much uncertainty around heterogeneities in susceptibility, fraction of cases that are truly asymptomatic, actual mixing levels within the US population, or even sub-populations, that including these in a model (while done by some) is hard to justify.

We have rewritten parts of the introduction to clarify what we are trying to illustrate!

Is Table 2 copied directly from CDC COVID-19 Response Team et al., “Severe Outcomes Among Patients with Coronavirus Disease 2019 (COVID-19) — United States, February 12–March 16, 2020,” *MMWR Morb. Mortal. Wkly. Rep.*, vol. 69, no. 12, pp. 343–346, Mar. 2020, doi: 10.15585/mmwr.mm6912e2?. If so could would this be better placed in the supplementary index?

This is copied directly from the cited reference. The authors believe this should be placed in the main paper as it is similar to Table 1 and supports the understanding of figure 2.

The authors mention homogeneity of their model population as a limitation, which I agree is a significant limitation of this approach. However, the discussion about reasons for potential variation in susceptibility focusses on the existence of cross-reactive T-cells from seasonal coronavirus exposure. Mentioning the age-dependence of susceptibility inferred in other modelling works (e.g. Davies et al., <https://www.nature.com/articles/s41591-020-0962-9>) is probably warranted here.

Agreed, added recommended reference to line 82 of the main paper.

In line 61 the use of broad is unclear: does broad refer to wide uncertainty in the value or widely varying between hosts (or both?).

Agreed the word is broad is unclear. The passage now reads: “The recovery and incubation period can be similarly difficult to estimate [e.g. 12] with wide uncertainty.”

Line 94 discussion of study in reference 31 is unclear, were participants tested longitudinally, otherwise how does this constrain the asymptomatic fraction?

The SIR model is used to model the spread of the disease to both symptomatic and asymptomatic individuals. Line 94 and reference 31 is used to show that the numbers in Table 2 were developed using only symptomatic individuals. Therefore, to deduce these numbers using the simple SIR model, one must account for the asymptomatic spread.

I think the method that has been adopted to infer cases has some limitations not mentioned. When inferring infections from deaths this is sensitive to the distribution of delays from infection to death, but the authors have assumed a fixed delay. A further limitation is that even if testing was abundant, changing restrictions may alter the age-distribution of cases and result in time-varying under-ascertainment.

The reviewer is correct. The way that we have done this is almost certainly an underestimate, both because of testing limitations and because fatality rates dropped as medical interventions became more effective. However, testing limitations were what they were, and there is no way around that. The impact that changes in medical interventions had on mortality rates is difficult to quantify in a time dependent manner and without that it cannot be factored into the model. We could consider the distribution for the delay between infection and death, but on average, this would not change the estimates. The bigger impact is definitely from the effects of limited testing and changing mortality rates. In order to address this, we have added the following to the discussion section:

“The way that infection rates were inferred from mortality data likely results in an underestimate. It is known that fatality rates decreased as medical interventions improved [28], [29]. However the temporal aspect is poorly quantified and was not considered as a result and this likely resulted in an underestimate of the infections. A fixed time from infection to death was also assumed while time to

mortality actually follows a distribution. On average, however, this should not strongly affect timing of predicted infections.”

Related to my major comment: from figure S4 it looks like the phase of the seasonality term has also been fixed and may plausibly vary by weeks-months. It could be worth including this effect in a sensitivity analysis.

From (R. A. Neher, R. Dyrdak, V. Druelle, E. B. Hodcroft, and J. Albert, “Potential impact of seasonal forcing on a SARS-CoV-2 pandemic,” *Swiss Med. Wkly.*, Mar. 2020, doi: 10.4414/sm.w.2020.20224.,) the variability in the magnitude of the seasonality far outweighs the variability in the phase of the seasonality effect. As seen below, from 2010 – 2019, the phase variability has a standard deviation of ~ 1 month for the time of maximum and minimum seasonality effect or 8% variability. The variability for the magnitude of the peak seasonality effect is 22%. Therefore, avoiding over fitting the range of uncertainty, the magnitude is greater than the phase seasonality.

Figure: Seasonality changes of Coronavirus from 2010 – 2019 from (R. A. Neher, R. Dyrdak, V. Druelle, E. B. Hodcroft, and J. Albert, “Potential impact of seasonal forcing on a SARS-CoV-2 pandemic,” *Swiss Med. Wkly.*, Mar. 2020, doi: 10.4414/sm.w.2020.20224.,).

We have also added to the discussion section the following:

“Seasonality was also assumed to change in magnitude, but not in timing. However, the data shows that the time when infection rates begin to rise varied by coronavirus strain by as much as \pm month. This could affect the best possible fit of the model to the data.”

SI line 19 - slightly confusing to refer to the birth rate as the growth rate when the later has another meaning in the paper.

SI material now reads:

“The infection rate is α (1/day) and β (1/day) is the recovery rate. In this work, the size of the population, $N(t)$ is a constant with no birth (i.e. $\mu=0$). “

SI line 100 - Mobility data doesn't track social interactions in households which are potential a large

source of transmission events, may be worth noting as another limitation.

Added the following sentence to SI material:

“A limitation to modeling the NPI is transmission within a household as its assumed households are not restricting mobility within the house and not wearing a mask at home.”

Figure S7 - serial interval shown in Table 1. Although the serial interval and generation time interval distributions should have the same mean, they are not the same so it is a bit confusing to use the terms interchangeably.

Caption for Figure S7 now reads:

“The figure shows the predicted range of possible outcomes for the US SARS-CoV-2 pandemic during 2020. The end of January, 2020, was taken as the baseline since that is the time period for which the **serial interval** time in Table 1, main paper, applies.”

Figure S7 - it may be my copy, but I can't easily make out the best fit in this figure, perhaps consider showing other fits as a shaded envelope?

See figure 3 of the main paper, we did exactly this.

Supplementary Note 2: this is for NPIs remaining at constant rate indefinitely? It could be worth clarifying this in the discussion.

SI Material now reads:

“The results from the full range of Monte Carlo simulations shown in Fig. 1 (left) can also be used to understand the effects of **NPI's at a constant level indefinitely**.”

Appendix C

Review of “Non-pharmaceutical interventions prevented ~250 million United States SARS-CoV-2 infections” by Duncan et al.

Thank you to the authors for the revised manuscript which addresses most of the concerns that I raised. I have some further suggestions that may improve the clarity of the results and interpretation:

- The newly included Figure S8 is a helpful addition, however for transparency and completeness should this be extended to include all sampled and selected parameters?
- The authors have included discussion of age as a likely source of heterogeneity in SARS-CoV-2 transmission as suggested. I am sympathetic to the argument that more complex models may not be justified given persistent uncertainties, however this argument for model choice doesn't come through as clearly in the manuscript and a more explicit statement when discussing sources of heterogeneity the introduction would be helpful.
- The authors included some statements regarding the limitations of their approach for linking cases to deaths, but I am not convinced that assuming that the time from infection to death is constant is obviously on average correct. The shape of the time to death distribution will also influence the back-calculated infection curve (particularly when there are sharp changes in transmission) and potentially introduce bias in the parameter sets selected (see, e.g. Keeling and Rohani, for a demonstration of the drawbacks of mis-specifying delay distributions).
- Line 164. I'm slightly confused by the justification for choosing the infectious period to be 7.5 days based shedding data rather than using the data on serial interval (which is around 5 days for many of the studies cited) when selecting the model parameters – this looks like it will fix the generation time in the model to be too long. Furthermore, neglecting the latent period and assuming exponentially distributed waiting times in the infectious class (equations S1-S4) can also bias estimates of R (Wearing, et al. 2005) and thus potentially also model calibration. Some sensitivity analysis here for the infectious period and/or model structure, or at least a discussion of these limitations, would seem useful.
- Issues of potential model mis-specification aside, by conditioning on parameters that have some uncertainty (e.g. the phase of the seasonal forcing, distribution of time to death etc) the estimate of uncertainty in the effect of NPIs will be underestimated. It may be worth stressing that $\sigma = 30$ million is likely an underestimate of the uncertainty in your estimate of averted infections.
- Figure 2 – I am unclear why hospitalizations feature here? My understanding is that it is just the testing and mortality data used to infer the total cases but if this is not the case it could be worth making this clearer in the methods. If the aim is to illustrate the divergence of epidemics with and without NPIs, should observed hospitalisations/deaths be plotted here too?
- Line 265 – please refer to figures/analysis to support the statement about the relative importance of seasonality compared to other drivers of changes in transmission. Plotting sampled and selected seasonality parameters may be useful here.
- Throughout it would be more common to refer to the serial interval rather than serial interval time, although this is probably largely convention.
- Line 44 – typo in underlying?

- Line 53 – compartmental models can account for inhomogeneous mixing via stratification, so perhaps qualifying with “*simple* compartmental models...” is probably warranted here.
- Line 62 (and elsewhere) – other coronaviruses circulating in humans seem to be usually referred to as seasonal human coronaviruses or common cold coronaviruses. “common coronaviruses” may be confusing here since SARS-CoV-2 is now also very common.
- Figure 1 (and other relevant figures, e.g. Figure 3) – please specify whether the y-axis is the incidence or prevalence. Does median indicate epidemic with median attack rate or some other measure?
- References to growth rate as a birth rate in figure caption within SI remains despite the changes within the text body.

References

Keeling, M. J., & Rohani, P. (2011). *Modeling infectious diseases in humans and animals*. Princeton university press.

Wearing, H. J., Rohani, P., & Keeling, M. J. (2005). Appropriate models for the management of infectious diseases. *PLoS medicine*, 2(7), e174.

Appendix D

The newly included Figure S8 is a helpful addition, however for transparency and completeness should this be extended to include all sampled and selected parameters?

Added a four figure to figure S7 for mobility, seasonality and mask wearing and combined effect with the best fit values added.

- The authors have included discussion of age as a likely source of heterogeneity in SARS-CoV-2 transmission as suggested. I am sympathetic to the argument that more complex models may not be justified given persistent uncertainties, however this argument for model choice doesn't come through as clearly in the manuscript and a more explicit statement when discussing sources of heterogeneity the introduction would be helpful.

Added the following text to lines 44 – 53 of the text.

Any degree of complexity can in principle be encompassed by expanding the number of compartments, and this is often done to consider different transmission rates among age groups and locations [e.g. 7]. However, if the underlying parameters are poorly understood, simpler models whose inputs have been measured can sometimes give better results [8]. As knowledge increases, the number of compartments can also be expanded. For SARS-Cov-2, even a 'minimal' SIR model requires understanding infection and recovery rates, mortality and hospitalization rates, and the effects of seasonality and non-pharmacological interventions. A wide range of estimates exist for these parameters, what affects them, and assumptions that underlay simple compartmental models in general. These will be discussed briefly to inform the ranges used.

The following text is also in the text in lines 319 – 320 that address this comment.

Robert May observed that it does not make sense to add complexity to a model if key parameters can only be guessed at [8].

And additional text in lines 325 -327 that address this comment.

John von Neumann is said to have wryly noted that with four parameters he could fit an elephant, with five he could make it wiggle its trunk [50]. However, the parameters here are neither free nor arbitrary.

- The authors included some statements regarding the limitations of their approach for linking cases to deaths, but I am not convinced that assuming that the time from infection to death is constant is obviously on average correct. The shape of the time to death distribution will also influence the back-calculated infection curve (particularly when there are sharp changes in transmission) and potentially introduce bias in the parameter sets selected (see, e.g. Keeling and Rohani, for a demonstration of the drawbacks of mis-specifying delay distributions).

Fair point and we have now tested this assumption. The following discussion is on lines 307 – 315 which addresses this comment with added description of additional simulations to explore the reviewers comment:

A fixed time from infection to death was assumed while time to mortality actually follows a distribution. On average, this is not expected to strongly affect the timing of predicted infections given both the inconsistencies in reporting for the date of mortality [45] and the 7-day moving average used here for infections. This assumption was tested by using a measured distribution for the time from diagnosis to death [Marschner, 2021], which we shifted by 5.1 days (the time from infection to symptoms), implemented without the 7-day moving average for infections. The predicted number of infections avoided changed to 248.2 million with a standard deviation of 28.1 with a negligible change in R^2 . However, a better approach would be to use a measured distribution for the time from infection to death.

- Line 164. I'm slightly confused by the justification for choosing the infectious period to be 7.5 days based shedding data rather than using the data on serial interval (which is around 5 days for many of the studies cited) when selecting the model parameters – this looks like it will fix the generation time in the model to be too long. Furthermore, neglecting the latent period and assuming exponentially distributed waiting times in the infectious class (equations S1-S4) can also bias estimates of R (Wearing. et al. 2005) and thus potentially also model calibration. Some sensitivity analysis here for the infectious period and/or model structure, or at least a discussion of these limitations, would seem useful.

The point of Table 1 is to point out the conflicting reports to estimate SIR parameters. We chose the distribution associated with a median of 7.5 days because it explicitly stated the distribution and it had a large distribution that encompassed the other study. We chose not to use the serial interval because the studies did not indicate the distribution type. These studies simply stated three points, two of which were the CI and then it was unknown if the third point was a mean, median or other statistical reference point. Given the three data points, it was obvious these distributions were highly right skewed and knowledge of the distribution type would be vital.

- Issues of potential model mis-specification aside, by conditioning on parameters that have some uncertainty (e.g. the phase of the seasonal forcing, distribution of time to death etc) the estimate of uncertainty in the effect of NPIs will be underestimated. It may be worth stressing that $\sigma = 30$ million is likely an underestimate of the uncertainty in your estimate of averted infections.

Added the precise mean and standard deviation to the text with the additional results of the extra simulations.

- Figure 2 – I am unclear why hospitalizations feature here? My understanding is that it is just the testing and mortality data used to infer the total cases but if this is not the case it could be worth making this clearer in the methods. If the aim is to illustrate the divergence of epidemics with and without NPIs, should observed hospitalisations/deaths be plotted here too?

Mortality data was used to infer total cases. Hospitalizations are a key figure during a pandemic, and Fig 2 (left) was used to show this information according to the simulations. To avoid confusion, we decided to move the left panel of Fig 2 to Supplementary Note 1 (Fig. S3). Consequently, the text referring to Fig 2 was changed to the following:

When all the simulations in Fig. 1 (left) are combined with the distribution of symptomatic people and the age dependent distributions for **mortality** rates shown in Table 2, the result can be used to determine the fraction of combinations which give a specific **mortality** rate for the US population. **This is shown in Fig. 2.** Here the output of these combinations was filtered to remove the simulated epidemics producing the 2.5% highest and lowest peak mortality rates. **Figure 2** then shows the range for the fraction of a population that **would be expected to die due to SARS-CoV-2** as a function of time along with the epidemic that gives 97.5th percentile, 50th, and 2.5th percentile **mortality** rates. All other possibilities between these limits sit within the shaded boundary. A heat map for their density is given in Supplementary Note 1. The same method is applied to the **hospitalization** rate, with the output of the combination of Monte Carlo simulation with the symptomatic distribution and the age specific **hospitalization** distributions filtered to remove the 2.5% highest and lowest **hospitalization** rates. This is shown in **Supplementary Note 1, Fig. S3.** The corresponding heat map is given in Supplementary Note 1. **These** results show that, in the absence of interventions, low impact epidemics are a small subset of the predicted outcomes.

- Line 265 – please refer to figures/analysis to support the statement about the relative importance of seasonality compared to other drivers of changes in transmission. Plotting sampled and selected seasonality parameters may be useful here.

Added the following text to lines 270.

The effect of seasonality on the progression of the epidemic is important and a factor in the resurgence of SARS-CoV-2 infections in the fall of 2020 **as depicted in Fig 3 (left).**

- Throughout it would be more common to refer to the serial interval rather than serial interval time, although this is probably largely convention.

- Line 44 – typo in underlying?

Fixed.

- Line 53 – compartmental models can account for inhomogeneous mixing via stratification, so perhaps qualifying with “*simple* compartmental models...” is probably warranted here.

Added Simple to text.

- Line 62 (and elsewhere) – other coronaviruses circulating in humans seem to be usually referred to as seasonal human coronaviruses or common cold coronaviruses. “common coronaviruses” may be confusing here since SARS-CoV-2 is now also very common.

Changed to “seasonal” human coronaviruses throughout paper.

- Figure 1 (and other relevant figures, e.g. Figure 3) – please specify whether the y-axis is the incidence or prevalence. Does median indicate epidemic with median attack rate

or some other measure?

Fixed

- References to growth rate as a birth rate in figure caption within SI remains despite the changes within the text body.

Fixed

References

Keeling, M. J., & Rohani, P. (2011). *Modeling infectious diseases in humans and animals*. Princeton university press.

Wearing, H. J., Rohani, P., & Keeling, M. J. (2005). Appropriate models for the management of infectious diseases. *PLoS medicine*, 2(7), e174.